# Dynamical evolution of a minor sudden stratospheric warming in the Southern Hemisphere in 2019

**Guangyu Liu[1], Toshihiko Hirooka[1], Nawo Eguchi[2] and Kirstin Krüger[3]**

[1]Department of Earth and Planetary Sciences, Kyushu University, Fukuoka, Japan
[2]Research Institute for Applied Mechanics, Kyushu University, Kasuga, Japan
[3]Department of Geosciences, University of Oslo, Oslo, Norway

*Correspondence to*: Nawo Eguchi (nawo@riam.kyushu-u.ac.jp), Guangyu Liu (liu.guangyu.465@m.kyushu-u.ac.jp)

**Abstract.**

A major strong sudden stratospheric warming (SSW) occurred in the Southern Hemisphere (SH) stratosphere in 2002 (hereafter referred to as SSW2002), which is one of the most unusual winter in the SH. Following several warmings, the polar vortex breakdown in midwinter. Eastward-travelling waves and their interaction with quasi-stationary planetary waves played an important role during this event. This study analyzes the Japanese 55-year Reanalysis (JRA-55) dataset to examine the SSW event that occurred in the SH in 2019 (hereafter referred to as SSW2019). In 2019, a rapid temperature increasing and decelerated westerly winds were observed at the polar cap, but since there was no reversal of westerly winds to easterly winds at 60°S in the middle to lower stratosphere, the SSW2019 is classified as a minor warming event.

The results show that quasi-stationary planetary waves of zonal wavenumber 1 developed during the SSW2019. The strong vertical component of the Eliassen–Palm flux with zonal wavenumber 1 is indicative of pronounced propagation of planetary waves to the stratosphere. The wave driving in September 2019 was larger than that of the major SSW event in 2002. Major SSWs tend to accompany preceding minor warmings, preconditioning, which changes the zonal flow that weaken the polar night jet as seen in SSW2002. A similar preconditioning was hardly observed in SSW2019. The strong wave driving in SSW2019 occurred in high latitudes. Waveguides (i.e., positive values of the refractive index squared) were found at high latitudes in the upper stratosphere during the warming period, which provided favorable conditions for quasi-stationary planetary waves to propagate upward and poleward.

## 1 Introduction

Sudden stratospheric warmings (SSWs) are extraordinary events that are regularly observed in the Arctic polar region during winter. Strong westerly winds associated with the polar vortex in the mid-to-high latitudes decelerate, and temperatures increase by several tens of Kelvins within a few days in the polar region during an SSW (Labitzke and van Loon, 1999; Andrews et al., 1987; Iida et al., 2014; Baldwin et al., 2021). Many studies have examined the underlying mechanisms of these events. The essential dynamical mechanism of the development of the SSWs is that enhanced quasi-stationary planetary waves propagate from the troposphere to the stratosphere and interact with the zonal mean flow (Matsuno, 1971). The occurrence of

SSWs is common in the Northern Hemisphere (NH) (Charlton and Polvani, 2007) but rare in the Southern Hemisphere (SH) (Roscoe et al., 2005; Naujokat and Roscoe, 2005). One of the reasons that SSWs rarely occur in the SH is the distribution of ocean–land and orography, which leads to smaller planetary wave amplitude in the SH (Andrews et al., 1987; Newman and Nash, 2005).

SSWs during mid-winter are classified as either major or minor warmings (Julian, 1967; Labitzke, 1968). Major warming events are defined by rapid temperature increases between 60° latitude and the Pole, and a breakdown of the polar vortex, where zonal-mean zonal winds at 10 hPa poleward of 60° latitude reverse from westerly to easterly. In contrast, minor warming events refer to high temperatures at the Pole without a reversal of zonal-mean zonal winds poleward of 60° latitude at 10 hPa. Moreover, major warmings can be classified as being of the "vortex-displacement" or "vortex-split" type depending on the structure of the polar vortex during the onset of the warming event (Charlton and Polvani, 2007).

It has been reported that minor SSWs characteristically precede major SSW as "preconditioning". The preceding minor SSWs are associated with planetary waves amplification of zonal wavenumber 1 concurrently with a minimum of the zonal wavenumber 2 (Labitzke1977; Labitzke1981; Bancalá et al., 2012). The "preconditioning" also changes in the zonal flow that weakens the polar night jet and thus favors the upward and poleward propagation of planetary waves (Andrews et al., 1987; Labitzke, 1981; Manney et al., 2009). Following the poleward propagating planetary waves, the polar vortices become vulnerable that lead to the causing major warmings. The presence of precondition is a necessary condition for a major SSW to occur but not a sufficient condition (Limpasuvan et al., 2004).

In the SH, minor warming events have occasionally been observed in mid-winter (i.e., Godson 1963; Labitzke and van Loon 1965; Barnett 1974; Al-Ajmi et all., 1985; Hirota et al., 1990; Shiotani et al., 1993), whilst only one major SSW event has been detected in 2002 (Roscoe et al., 2005; Naujokat and Roscoe, 2005). Before the onset of SSW2002, a sequence of amplified planetary-wave activity was observed, which played an important role in weakening the polar night jet (PNJ) (Krüger et al., 2005). Then, the polar vortex broke down in September and split into two. The strong eastward-traveling waves, consisting primarily of planetary waves of zonal wavenumber 2, led to wave-mean flow interactions that weakened the PNJ, whilst the amplified quasi-stationary waves caused the disruption of the polar vortex and abruptly increased the polar temperature. The SSW2002 was classified as a major warming event of the "vortex-split" type applying the criteria of Charlton and Polvani (2007). The SSW2002 in the SH also significantly impacted the interannual variability of the Antarctic ozone hole (Weber et al., 2003). The warm air and particularly strong wave activity during SSW2002 disrupted the depletion of ozone over Antarctica, leading to the smallest ozone hole since 1988 (Allen et al., 2003; Newman and Nash, 2005; Stolarski et al., 2005).

In September 2019, a strong SSW (SSW2019) occurred in the SH (Yamazaki et al., 2020; Hendon et al., 2019; Eswaraiah et al., 2020). Rao et al. (2020) investigated the predictability of an SSW event that occurred in the SH in 2019 based on subseasonal to seasonal (S2S) models and identified favorable conditions such as easterly equatorial quasi-biennial oscillation (QBO) winds at 10 hPa, solar minimum, and positive Indian Ocean Dipole (IOD) sea surface temperatures that may have led to its occurrence. Following SSW2019, a significant reduction of the ozone hole area was detected during the peak ozone depletion period based on the Aura Microwave Limb Sounder (MLS) of Aura satellite and Global Earth Observing System model simulations (Wargan et al., 2020). Safieddine et al. (2020) showed that the total ozone poleward of 45°S increased during September to November 2019 using the Infrared Atmospheric Sounding Interferometer. Shen et al. (2020) suggested the original of planetary waves of zonal wavenumber 1 in the troposphere and implied a potential but unlikely to be a direct cause of the tropical easterly phase of the QBO in the upper stratosphere in facilitating the weakening of polar vortex. Quasi 6-day waves in the mesospheric winds were detected during the SSW2019 in low latitude, which is attributed to instability in the SH high latitude mesosphere (Lee et al., 2021).

The purpose of this study is to investigate the dynamical evolution of the SSW2019 and compare it with the SSW2002 event in the SH. The data and analysis methods are described in Section 2, followed by a discussion of the evolution and dynamical features of SSW2019 in Section 3. We describe the features of SSW2019 in Section 3 and discuss the effect of reflective index squared of stationary planetary waves in Section 4. We present summary and conclusion in Section 5.

## 2 Data and Analysis Methods

### 2.1 The JRA-55 reanalysis data

In this paper we use horizontal winds, temperature, and geopotential height from the Japanese 55-year Reanalysis (JRA-55) dataset provided by the Japan Meteorological Agency. Because major SSW is not observed in the SH before 2002, the analysis period is from 1979 to 2019 since there are limited observation at high latitudes in the SH before 1979 and the grid resolution is $1.25° \times 1.25°$ in the longitude–latitude directions. We use daily averages of the original 6-hourly data. The climatological mean is calculated over 41 years (1979-2019). Details of the data are described in Kobayashi et al. (2015). The Stratosphere-troposphere Processes And their Role in Climate (SPARC) Reanalysis Intercomparison Project (S-RIP) gives an evaluation of individual reanalysis datasets (Fujiwara et al., 2017).

### 2.2 Analysis methods

To analyze the wave-mean flow interaction, we consider planetary wave with zonal wavenumbers from 1 to 3 (i.e., planetary scales) based on the Transformed Eulerian Mean equations. We employ the Eliassen–Palm flux (E–P) to study the effect of the wave forcing on the zonal mean circulation. The vector of the E–P flux represents the direction of wave energy propagation in the zonal-mean circulation system. Moreover, the total wave forcing can be represented by the divergence

(convergence) of the E–P flux, which is related to the acceleration (deceleration) of the westerly zonal-mean circulation (Andrews and McIntyre, 1976; Andrews et al., 1987). The E–P flux methodology in the quasi-geostrophic form is given by:

$$\boldsymbol{F} = \{F_y, F_z\} = \left\{ -\rho a \cos\theta \, \overline{u'v'}, \rho a \cos\theta f \frac{\overline{v'\phi_z'}}{N^2} \right\}$$

(1)

where $F_y$ and $F_z$ represent the meridional and vertical components of the E–P flux, respectively. The zonal and meridional winds are denoted by $u$ and $v$, respectively, and the prime denotes small perturbations to zonal mean flow. In this study, Fourier-analyzed amplitude of planetary waves of different zonal wavenumbers are calculated as well as the wave components of the E–P flux and its divergence. The radius of the earth, the buoyancy frequency, density, latitude, vertical gradient of geopotential height, and the Coriolis parameter are represented by $a, N, \rho, \theta, \phi_z$, and $f$, respectively.

To study wave propagation, the distribution of the refractive index squared is analyzed based on

$$n_0^2 = \frac{\overline{q}_\phi}{a\overline{u}} - \frac{f^2}{4N^2H^2}$$

(2)

where $n_0^2$ is the refractive index squared; $\overline{q}_\phi$ is the meridional gradient of zonal mean potential vorticity; $\overline{u}$ denotes zonal-mean zonal winds; and $H$ is the scale height. For more details, see Andrews et al. (1987). Waves can propagate in regions of positive refractive index squared and are evanescent in the negative regions.

**3 Results**

**3.1 Overview of SSW2019**

We present an overview of the SSW2019 and SSW2002. Figure 1 shows the time-height cross-sections of the zonal-mean temperature difference, $\Delta T$, between 60°S and the South Pole, and the zonal-mean zonal winds at 60°S in 2019, and 2002 for comparison. Firstly, intermittent warmings (positive $\Delta T$) occur in the upper stratosphere (~5 to 1 hPa) from mid-August to mid-September 2002. A clearly visible warming (a positive $\Delta T$) emanates down to 100 hPa in late-September. Secondly, intermittent warmings lead to weakening of zonal-mean zonal winds in the upper stratosphere. Periodic weakening and strengthening of westerlies appear from mid-August to mid-September. A reversal of the zonal-mean zonal winds from westerlies to easterlies reaching down below 10 hPa appears at 60°S in late-September, which fulfills the World Meteorological Organization (WMO) criterion of a major SSW (e.g., WMO, 1978). Easterlies appear again in late-October after the westerlies. The observational description of the SSW2002 has been well reported by Krüger et al. (2005).

In 2019, regular oscillation of warmings (positive $\Delta T$) occur in the upper stratosphere (~5 to 1 hPa) from June to the first half of July. Except for a short warming period in the upper stratosphere in mid-August, temperatures over the South Pole are lower than those at 60°S until late August. After a couple of warming pulses from late August to early September, conspicuous warming pulses (positive $\Delta T$) occur in the upper stratosphere at the South Pole, which correspond to the SSW occurrence. The positive $\Delta T$ with values of about 15K propagates downward to the middle stratosphere (~20 hPa) until late October. Zonal-mean zonal winds are regularly weakened in the upper stratosphere correspond to the warmings from June to

the first half of July. From late August to early September, there are two substantial weakening periods of the PNJ from values exceeding 80 m s$^{-1}$ to 20 m s$^{-1}$ in the upper stratosphere (~5 to 1 hPa). A reversal of the zonal-mean zonal winds from westerlies to easterlies occurs in mid-September in the upper stratosphere. Subsequently, weak westerlies occur in the upper stratosphere, which lasts until the mid-October. Easterly winds occur again in the upper stratosphere in second half of October, leading to the gradual transition to the summer circulation. Since the reversal of zonal-mean zonal winds from westerlies to easterlies does not occur at 10 hPa and 60°S, SSW2019 is classified as a minor SSW.

**3.2 Synoptic evolution**

Figure 2 shows the synoptic evolution of temperature and geopotential height at 10 hPa on selected days in 2019. During the period 25–27 August, the cold polar vortex locates over the South Pole. It is partly surrounded by an anticyclone, with warm air on the edge of the polar vortex near southern Africa. During the period 28-30 August, the temperature around the Pole begins increasing and the anticyclone in the south of Australia begins to develop. From 31 August to 2 September, the high temperature region becomes larger, whilst the low temperature region shifts off the South Pole. From 3–5 September, the temperatures decrease at the edge of the polar vortex while the vortex itself weakens further. The low temperature region shifts off the centers of the vortices, indicating baroclinic conditions. Between 6 and 8 September, the high temperature stretches poleward, almost reaching the South Pole. The warming culminates on 11 September with a weakening of the polar vortex. The anticyclone also develops strongly during this period. After the peak warming, the warm air remains over the South Pole from 12 to 20 September and the anticyclone moves to the southwest of Australia.

The temperatures in the middle stratosphere are higher than average during the period from late August to September 2019. Figure 3a shows zonal-mean temperatures at 90°S and 10 hPa from 1 June to 31 October. The climatological temperature reaches its minimum around June and the interannual variability is relatively small at that time. After that, the temperature gradually increases and the interannual variability becomes larger, especially from September to October. In 2019, the temperature is close to the average until mid-August. Several warmings occur in late August, with pronounced warmings on 31 August and 11 September. The temperature increase ($\Delta T$) between 31 August and 11 September is ~40 K (hereafter referred to as the warming period). After the large temperature increase, a slight decrease occurs, but the high temperatures last for around one more week. Finally, the temperature attains a peak value of ~275 K on 19 September, which is about 10 days earlier than in 2002 (green line). The magnitude of the warming peak over the South Pole in September 2019 is well outside the standard deviation of the climatological temperature.

Figure 3b shows the zonal-mean zonal winds at 60°S and 10 hPa. The climatological zonal-mean zonal wind peaks in August and decreases afterward, with large interannual variability. In 2019, a pronounced deceleration of the westerly winds to ~61 m s$^{-1}$ occurs on 31 August, in accordance with the warming in late August. The westerly wind reaches a value of ~26 m s$^{-1}$ on 11 September, coinciding with a warming peak in the temperature. The decrease in the magnitude of the wind ($\Delta U$)

is ~35 m s$^{-1}$ during the warming period. The deceleration continues until mid-September, with the minimum westerly winds occurring on 17 September (~11 m s$^{-1}$). The magnitude of the weakening is ~50 m s$^{-1}$ between 31 August and 17 September 2019. In 2002, the zonal-mean zonal winds reverse to easterly winds on 27 September (~20 m s$^{-1}$), resulting in a difference of ~72 m s$^{-1}$ from 24 August, when the first warming pulse occurs. Like the temperature evolution in 2019, the zonal-mean zonal winds are well outside the standard deviation of the climatology during September. There is no occurrence of the zonal-mean zonal wind reversal in 2019, which is one of the differences compared with SSW2002.

**3.3 Dynamical evolution**

The quasi-stationary planetary waves of zonal wavenumber 1 (PW1) plays an important role in the dynamical evolution of SSW2019. Figure 4 shows planetary wave amplitudes of PW1 and PW2 at 60°S and 10 hPa (top) and upward E– P fluxes for PW1–3 at 60°S and 50 hPa (bottom) for 2019 (left) and 2002 (right). The largest amplitude of PW1 exceeds 2000 m on 8 September (~2137 m). Yamazaki et al. (2020) reports that this is the highest value of amplitude of PW1 that has been observed since August 2004 by Aura MLS in the SH. Large values of amplitude of PW1 also be found in late August and the first half of September. These large amplifications of PW1 disturb the polar vortex, leading to a weakening of the PNJ (Eswaraiah et al., 2020). The large growth of PW1 could be associated with the easterly phase of the quasi-biennial oscillation in the SH tropics (Shen et al., 2020; Rao et al., 2020). In comparison with the predominant role of PW1, PW2 in SSW2002, PW2 appears to be less dominant during the warming period in SSW2019. Furthermore, the eastward-traveling PW2 presents around 31 July, 10 August, and 20 September 2019 at 10 hPa (not shown) but is not as pronounced as in 2002 (Krüger et al., 2005).

The vertical component of the E–P flux (hereafter EPFz) is a useful diagnostic for evaluating the vertical propagation of planetary waves into the stratosphere (e.g., Harada and Hirooka, 2017). Here we decompose the EPFz into components of zonal wavenumbers 1–3 to gain a deeper understanding of individual contributions of the planetary wave during SSW2019 and SSW2002 in the stratosphere. In Fig. 4b, d, the total EPFz at 50 hPa for all wavenumbers is shown by gray shading, along with the individual contributions from PW1–3 by colored lines. The total EPFz in 2019 indicates a high activity of planetary waves propagating into the stratosphere beginning in late August and attaining peak values in the first half of September, in accordance with the increasing temperatures at the South Pole and the weakening westerly winds. In addition, the peak value of the total EPFz in 2019 does not surpass that in 2002. Furthermore, the contribution of PW1 is considerably large in 2019. SSW2019 is characterized by the large growth of PW1 activity that disturbed the polar vortex during the warming period. In contrast to the role of PW1, PW2, and PW3 in SSW2002, PW2 and PW3 appear to be less pronounced during the warming period in SSW2019.

Figure 5 shows the latitude-height cross-sections of zonal-mean zonal winds on several selected days. During the period 25–27 August, the PNJ core located in about 60°S and in the range from 5 to 1 hPa. As mentioned above, large-

amplitude wave occurred from late August to the first half of September. During the period 28 to 30 August, anticyclone and temperatures around the Pole begin developing and increasing as seen in Figure 2. During that period the core of the PNJ is also considerably weakened from to about 70 m s$^{-1}$ from a value exceeding 90 m s$^{-1}$ between 25 and 27 August. Due to the large wave activity starting in late August, a substantial deceleration of the PNJ takes place from 31 August to 11 September. Except for a slight strengthening of the PNJ from 3 to 5 September, the PNJ is continually weakened during warming period, in line with the large temperature increase observed in Figure 3. Furthermore, the core of the PNJ propagates downward and the axis shift poleward in the stratosphere during SSW2019. After the substantial deceleration of the PNJ, westerly winds remain relatively weak from 12 to 20 September and the characterized poleward shift of the PNJ axis exists below 10 hPa. The deceleration of the PNJ from 12 to 20 September is in accordance with the warming over the South Pole observed in Figure 2. The poleward shift of the westerly PNJ indicating the baroclinic conditions as seen in Figure 2.

We have examined the evolution of planetary wave from the troposphere to the stratosphere in terms of the E–P flux. Figure 6 shows the time-height cross-sections of E–P flux vectors and the divergence for PW1 and PW2 at 60°S. E-P flux vectors pointing to the right and up directions represent poleward and upward, respectively. From Fig. 6a, in addition to a pronounced upward and poleward propagation of PW 1, strong convergence of the E–P flux could be found in the upper stratosphere in late August and the first half of September. The strong convergence in the upper stratosphere leads to the sudden warming by weakening the polar vortex. In contrast to the PW1, the PW2 is relatively weak during the warming period (Fig. 6b). This suggests that strong upward and poleward propagation of PW1 and strong convergence played an important role in triggering the SSW2019.

The evolution of planetary waves for SSW2002 has been well documented by Baldwin et al. (2003, their Fig. 6). Our Figure 6c, d confirms that both PW1 and PW2 periodically strengthens and propagates from the troposphere to the stratosphere by late September. Strong convergence of the E–P flux appears intermittently in the upper stratosphere for both PW1 and PW2 by September. This suggests that the PNJ and polar vortex were weakened by the intermittently strong planetary waves, preconditioning the stratosphere before the occurrence of SSW2002 in late September as mentioned earlier (Krüger et al., 2005). Subsequently, the polar vortex broke down due to the large planetary waves in late September, which resulted in the reversal of the zonal-mean zonal wind at 60°S and 10 hPa, as shown in Figure 3b.

Figure 7 shows the latitude-height cross-sections of the E–P flux and the E–P flux divergence (convergence), which is related to the acceleration (deceleration) of the zonal-mean zonal winds, on the same selected days as for Figure 5. Pulses of strong wave forcing are observed in the stratosphere at high latitudes from late August to the first half of September 2019. From 28 to 30 August 2019, the planetary waves strongly propagate upward and poleward from 60°S. Strong convergence is observed in the upper stratosphere, which corresponds to the strongly amplified planetary waves that lead to the deceleration of the PNJ mentioned above. From 31 August to 11 September, the waves propagate upward and equatorward, and the E–P

flux converges in the upper stratosphere extratropic. During the period 6–8 September, a second maximum in the E–P flux convergence occurs, with wave propagation from the troposphere to the upper stratosphere at around 60°S. This convergence contributes to the occurrence of SSW2019 by decelerating the PNJ and warming the polar cap. Following the warming period with considerably strong planetary waves, regions of the E–P flux convergence remain in the high latitudes around 10 hPa until 20 September. The long duration of the E–P flux convergence corresponds to the continuously warming and weakening PNJ shown in Figures 2 and 5.

The propagation of planetary waves in the stratosphere play an important role in triggering SSW2019. As mentioned previously, strong propagation of planetary waves took place in high latitudes from late August to the first half of September 2019. To understand the strong propagation of the planetary waves from the troposphere to the stratosphere in high latitudes, we have examined the refractive index squared that conducive to planetary wave propagating in the stratosphere (Newman and Nash, 2005). Figure 8 shows the meridional cross-sections of the refractive index squared $n_0^2$ before, during, and after the warming period. From 31 August to 11 September 2019 (during the SSW2019), a wide waveguide (i.e., positive $n_0^2$) form from the troposphere to the stratosphere around 60°S. As planetary wave packets tend to propagate in regions with a large positive value of $n_0^2$, planetary waves are allowed to propagate upward into the stratosphere through this waveguide. Because the existence of the waveguide during the warming period, the PNJ reduces to about 55 m s$^{-1}$ from the period before the warming. On the other hand, the waveguide forms toward the polar stratosphere with height during the SSW2019. This poleward waveguide provides the poleward planetary wave propagation as mentioned previously. After the warming period, the persistent waveguide in high latitudes present until 20 September. This waveguide allows the continuous propagation of planetary waves from the troposphere to the stratosphere to continually warm the polar region by weakening the PNJ.

As shown in Figures 4 and 7, there are large amplifications of planetary waves and strong wave driving represented by the convergence of the E–P flux in the stratosphere in September 2019. Because the upward propagation of planetary waves from the troposphere to the stratosphere develop from July and peaks in September (Lim et al., 2021). To compare the total planetary wave forcing on the zonal flow in the analysis period as well as contributions from the wave forcing of zonal wavenumber 1 and 2, we have examined the divergence (convergence) of the E-P flux that is related to the acceleration (deceleration) of the westerly zonal-mean circulation. Figure 9 shows time series of the divergence (convergence) of the E–P flux for PW1 and PW2 between 30°S and 90°S at 10 hPa in September. Firstly, it is evident that the magnitude of the convergences of the E–P flux is larger in 2019 than in any other year within the past 18 years, which means strong westerly deceleration in 2019 than other years. In addition, SSW2019 is predominantly driven by planetary wave forcing of zonal wavenumber 1. Also, the magnitude of the convergence (westerly deceleration) of the E–P flux in 2002 is the second largest within the analysis period. Moreover, in contrast to the predominant of PW1 in 2019, both PW1 and PW2 contribute the wave forcing on the zonal flow in 2002.

**4 Discussion**

Even though SSW2019 did not fulfil the criterion of a major SSW, the large increasing temperature in high latitudes still has a significant impact on the stratosphere. Due to the remarkable increased polar stratospheric temperature, slowing down catalytic chemical reaction on polar stratospheric clouds that suppress the formation of the Antarctic ozone hole in austral spring. Indeed, a diminished Antarctic ozone hole area is observed in 2019 (Wargan et al., 2020; Safieddine et al. (2020). As described in previous section, SSW2019 resulted from the pronounced planetary wave forcing especially the contribution from zonal wavenumber 1. In this section, we consider such unusual features in SSW2019 and compare it with SSW2002.

As mentioned above, one striking difference between the unusual major SSW2002 and minor SSW2019 in the SH is that the zonal-mean zonal winds did not reverse to easterly winds in 2019. Preconditioning is considered as a characteristic of major SSWs (Labizke, 1981) and many studies have demonstrated the importance of preconditioning in SSW2002 (Allen et al., 2003; Baldwin et al., 2003; Newman and Nash, 2005). Krüger et al. (2005) highlighted the importance of the interaction of eastward-traveling PW2 with quasi-stationary PW1, which considerably weakened the PNJ before the major SSW. However, the quasi-stationary PW1 in 2019 is not amplified (nor large) before late August as in 2002 except for June (Figure 4). In addition, the eastward-traveling PW2 is less active and pronounced before the occurrence of SSW in 2019 (not shown). As necessary condition for a major SSW, preconditioning before the warming and interaction between the eastward-travelling PW2 with the quasi-stationary PW1 are not pronounced in 2019.

The SSW2019 occurs when the PNJ still has strong westerly winds, which is one of the reasons that a reversal of the zonal-mean zonal winds does not occur at 10 hPa and below. Except for the periods in June and mid-August, westerly winds are stronger than or close to the normal throughout austral winter in 2019. In addition to the strong westerly winds, unlike the periodic weakening and strengthening of the zonal-mean zonal winds before the SSW occurs in 2002, the strength of the PNJ is less disturbed in 2019. Because the strong convergence of the E–P flux in the high latitudes, westerly winds are decreased by at least about 50 m s$^{-1}$ by mid-September (see Figure 3). Similar or even smaller magnitudes of deceleration can result in a reversal of the zonal-mean zonal winds, as was observed for the major SSW in the NH winter of 2018/2019 (Rao et al., 2020; Wargan et al., 2020).

Because the strong planetary wave forcing in high latitude during the warming period, a substantial decreasing of the westerly winds is found in the stratospheric high latitude. In contrast with the reversal from westerly winds to easterly winds in the stratosphere in the SSW2002 (Newman and Nash, 2005). Following with the westerly winds decreasing, a characterized poleward shift of the PNJ axis is found in the SSW2019. The poleward shift of the PNJ axis suggests baroclinic conditions in the stratosphere, which is considered as be attributed to exiting the planetary waves in the stratosphere as studied by Yamazaki et al. (2020). The core of the PNJ is close to the Pole in SSW2002, which suggests that the refractive index squared also shifts

toward the South Pole (Newman and Nash, 2005). The poleward shift of the PNJ axis with height in the SSW2019 also impacts the planetary waves propagation into the stratosphere as will be discussed afterward.

Newman and Nash (2005) suggested that the refractive index squared facilitates the propagation of planetary wave in the stratosphere in SSW2002. During SSW2019, a wide waveguide is found from the troposphere to the stratosphere in high latitudes. Because planetary waves tend to propagate in large positive values of the refractive index squared, planetary waves are considered to propagate upward through this waveguide. This is consistent with the strong upward propagating wave as seen in Figures 6 and 7. On the other hand, the waveguide shifts poleward with height during SSW2019. This is considered as be attributed to the baroclinic condition in the stratosphere in 2019. It suggests that the poleward waveguide facilitates planetary wave propagation to the polar region, which produces to the warming in the polar region. Newman and Nash (2005) suggested that as the core of PNJ shifts to the Pole the refractive index squared also shifts toward the Pole in 2002. In SSW2019 due to the baroclinic condition the PNJ and the waveguide shift toward the Pole with height during the warming. The wide poleward shift of waveguide is considered to facilitate the propagation of planetary waves to the stratosphere that produce SSW2019.

The values of total planetary wave forcing and PW1 are the largest in September 2019 during the analysis period. The second largest values of the total forcing in September are found in 2002 when the major SSW occurs. For both 2019 and 2002, the warmings and strong westerly decelerations are attributed to the large wave forcing. In 2019, the wide waveguide at high latitude facilitates the planetary wave propagation from the troposphere to the stratosphere as shown in Figure 8. Beside the dominant role of PW1 in weakening the PNJ, PW2 is relatively weak in SSW2019. As mentioned previously, the strong waves of zonal wavenumber 1 to 3 before the warming lead to a weakening of the PNJ, the preconditioning, plays an important role in the occurrence of SSW2002, which is considered as favor the propagation of PW2 as studied by Krüger et al. (2005). Even though the strongest wave forcing and PW1 are observed, unpronounced preconditioning before the warming and insufficient presence of other zonal wavenumbers are two reasons that the major SSW did not happen in 2019. Shen et al. (2020) suggested the persistent of anomalous convection in the troposphere over the South Pacific as the source of the PW1. On the other hand, Yamazaki et al. (2020) suggested that the source of the pronounced planetary waves was attributed to the barotropic condition in the stratosphere. Hence, further studies are still required on the occurrence of preconditioning of 2019 and the lack of planetary waves of other zonal wavenumbers than zonal wavenumber 1 in 2019.

**5 Summary and Conclusion**

In this study, the evolution of the Sudden Stratospheric Warming 2019 (SSW2019) in the Southern Hemisphere (SH) was analyzed using the JRA-55 meteorological reanalysis. Large increased temperatures and decelerated westerly winds were observed in the southern polar region in September 2019. Even though large increasing temperatures happened, a reversal from westerly winds to easterly winds did not take place at 60°S and 10 hPa, SSW2019 in the SH cannot be classified as a major SSW but as a minor SSW.

Temperatures increased strongly in the first part of September following a couple of warmings in late August. The temperatures at the South Pole were well above the climatological average and out of the standard deviation during most of September. In accordance with the pronounced warming at the Pole, the westerly winds decelerated significantly in the stratosphere at high latitude from late August. The decreased westerly winds were well below the average and out of the standard deviation during September. Although a reversal of zonal-mean zonal winds from westerlies to easterlies was observed in the upper stratosphere in early September, this reversal did not reach down to 10 hPa at 60°S.

The present study has shown that there was a pronounced amplification of the quasi-stationary PW1 during SSW2019. The propagation of planetary waves into the stratosphere was investigated using the vertical component of the E–P flux. Strong planetary waves with a large contribution of PW1 propagated into the stratosphere in high latitude. Strong planetary wave driving, represented by the convergence of E–P flux, occurred in the upper stratosphere during SSW2019, which led to the weakening of the PNJ and warming the Pole. In contrast to the regular occurrence of the eastward-traveling PW2 during SSW2002, the quasi-stationary PW1 played a dominant role in SSW2019. By studying the interannual variability of the wave forcing in September, we showed that the total wave forcing and the contribution of PW1 was larger in 2019 than in any other year during the analysis period (1979–2019).

As the large-amplitude wave occurred from late August to the first half of September, a substantial deceleration of the PNJ took place during the warming period. In addition, the core of the PNJ propagated downward and poleward shift of the PNJ axis existed during the warming period and last until late September. The poleward shift of the westerly PNJ indicated the baroclinic conditions in the stratosphere in 2019. Large planetary wave forcing represented by the convergence of the E-P flux were found in the stratosphere during the warming period in high latitude. The large planetary wave forcing decelerated the westerly winds and produced the warming in the high latitude in 2019.

The refractive index squared analysis showed that during SSW2019, planetary waves propagated upward to the stratosphere through an open waveguide in the high latitudes. We found that a wide waveguide appeared in high latitudes from the lower to the upper stratosphere during SSW2019, which allowed planetary waves to propagate through the stratosphere. In addition, the waveguide is formed to be inclined toward the Pole with height, which facilitates the poleward propagation of planetary waves. Moreover, because the waveguide existed after the pronounced warming, it allowed planetary waves to propagate upward to continually weaken the PNJ. This revealed that strong and long-lasting quasi-stationary PW1 propagated to the stratosphere during SSW2019.

**Data availability**

The JRA-55 data set used in this paper is available on the JMA Data Dissemination System (https://jra.kishou.go.jp/JRA-55/index_en.html).

**Author contribution**

TH, NE, and KK designed the study, provided guidance and in the interpretation of the results, and reviewed the manuscript.

GL performed the analysis and wrote the manuscript with contributions from TH, NE, and KK.

**Competing interests**

The authors declare that they have no conflicts of interest.

**Acknowledgments**

This study was partially supported by Grant-in-Aid for "2019 Initiative for Realizing Diversity in the Research Environment" through the "Diversity and Super Global Training Program for Female and Young Faculty (SENTAN-Q)", Kyushu University from MEXT and by JSPS KAKENHI Grant numbers JP18H01280, JP18H01270, and JP20H01973. The GFD-DENNOU Library was used for graphical output.

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

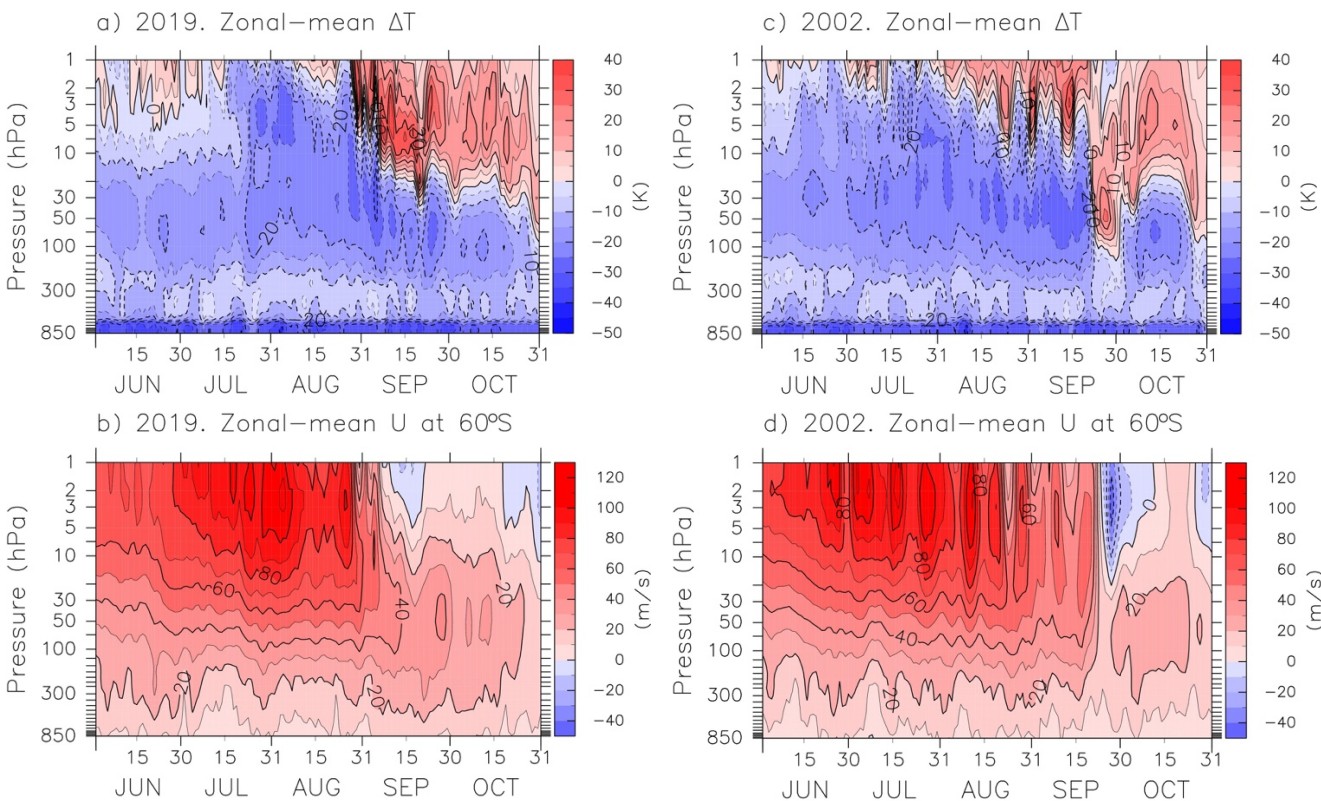

2    Figure 1. Time–height cross-sections of the temperature difference [K] between 60°S and the South Pole (a,c) and the zonal-

3    mean zonal wind [m s$^{-1}$] at 60°S (b,d) from 1 June to 31 October for 2019 (left) and 2002 (right). The contour intervals are 5

4    K for temperature and 10 m s$^{-1}$ for zonal wind, respectively.

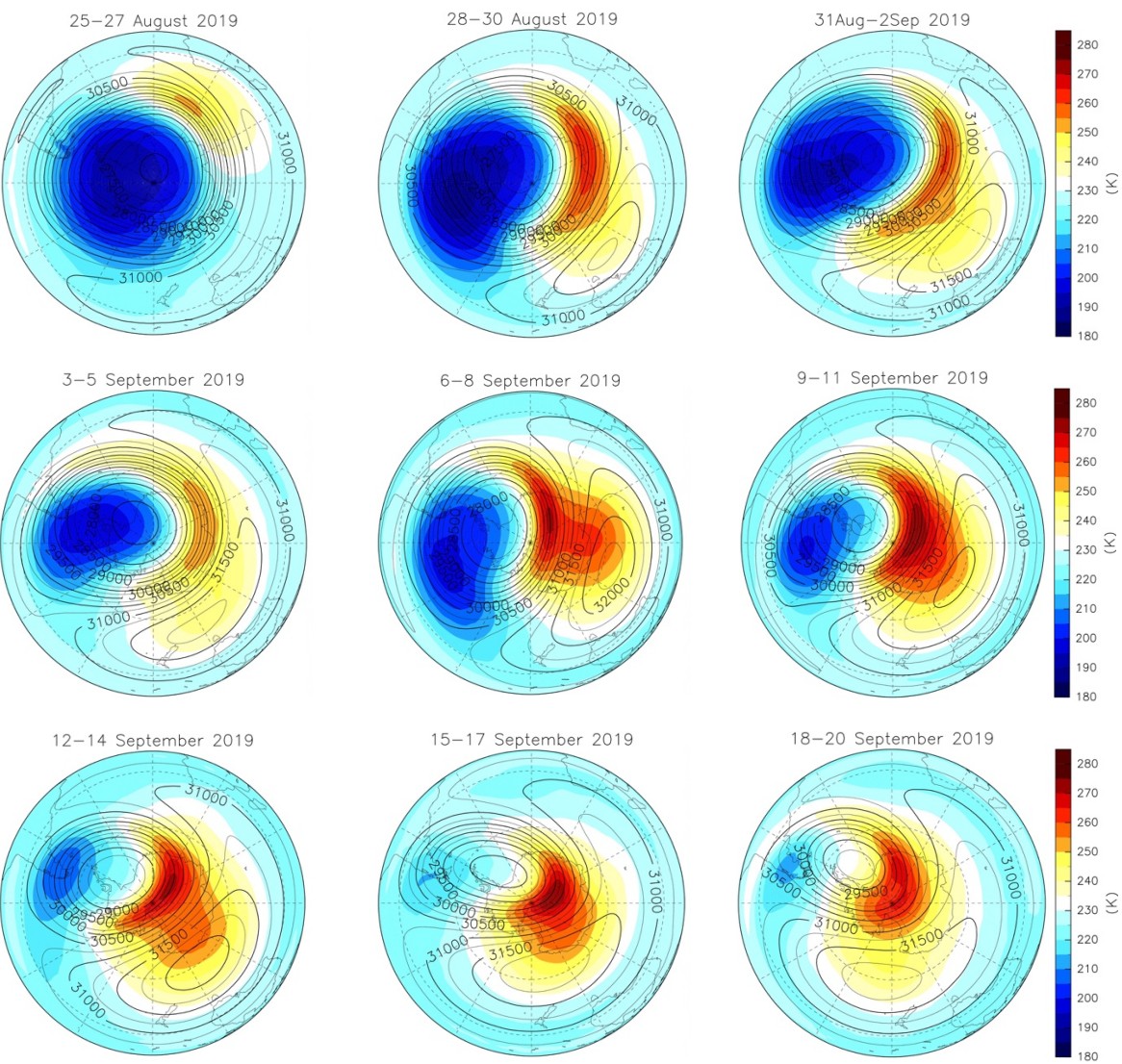

2  Figure 2. Polar stereographic map of temperatures [K] (colour shading) and geopotential heights [m] (contours) in the Southern

3  Hemisphere at 10 hPa for successive 3-day mean from 25 August to 20 September 2019. Contour intervals are 250 m.

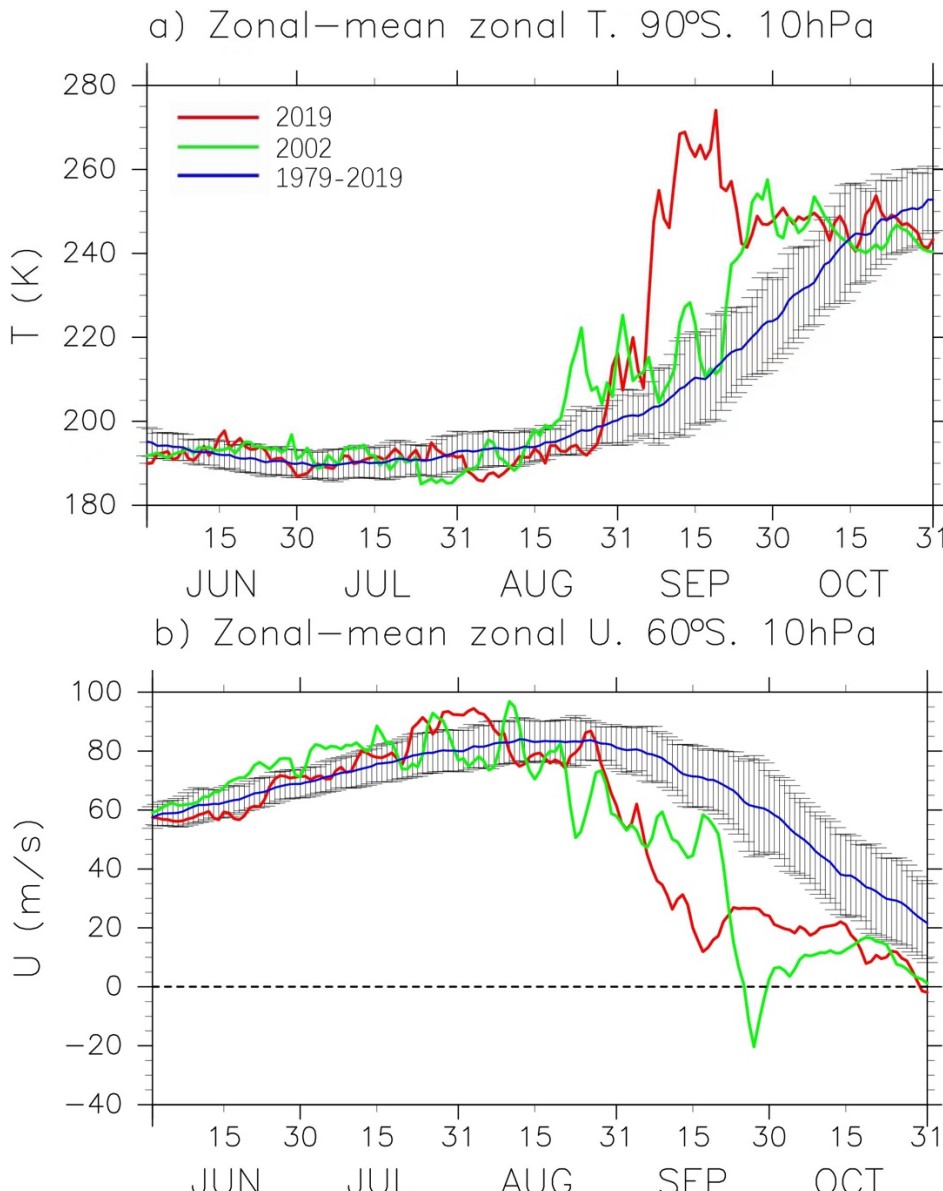

2    Figure 3. Time series of temperature [K] at 90°S and 10 hPa (a) and zonal-mean zonal wind [m s⁻¹] at 60°S and 10hPa, (b)

3    from 1 June to 31 October. Climatological values (blue) from 2002 (green) to 2019 (red) are represented with one standard

4    deviation shown by error bars.

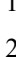

Figure 4. Time series of planetary waves amplitudes [m] at 10 hPa and 60°S (a,c) and the vertical component of the E-P flux [× $10^4$ kg s$^{-2}$] at 60°S and 50 hPa (b,d) from 1 June to 31 October for 2019 (left) and 2002 (right). The red, blue, and green lines denote the zonal wavenumbers 1, 2, and 3, respectively. In the bottom panels, grey shadings show the vertical component of the E-P flux of all wavenumbers.

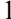

Figure 5. Latitude–height cross-sections of zonal-mean zonal wind [m s⁻¹] averaged every three days from 25 August to 20 September 2019. Contour intervals are 5 m s⁻¹.

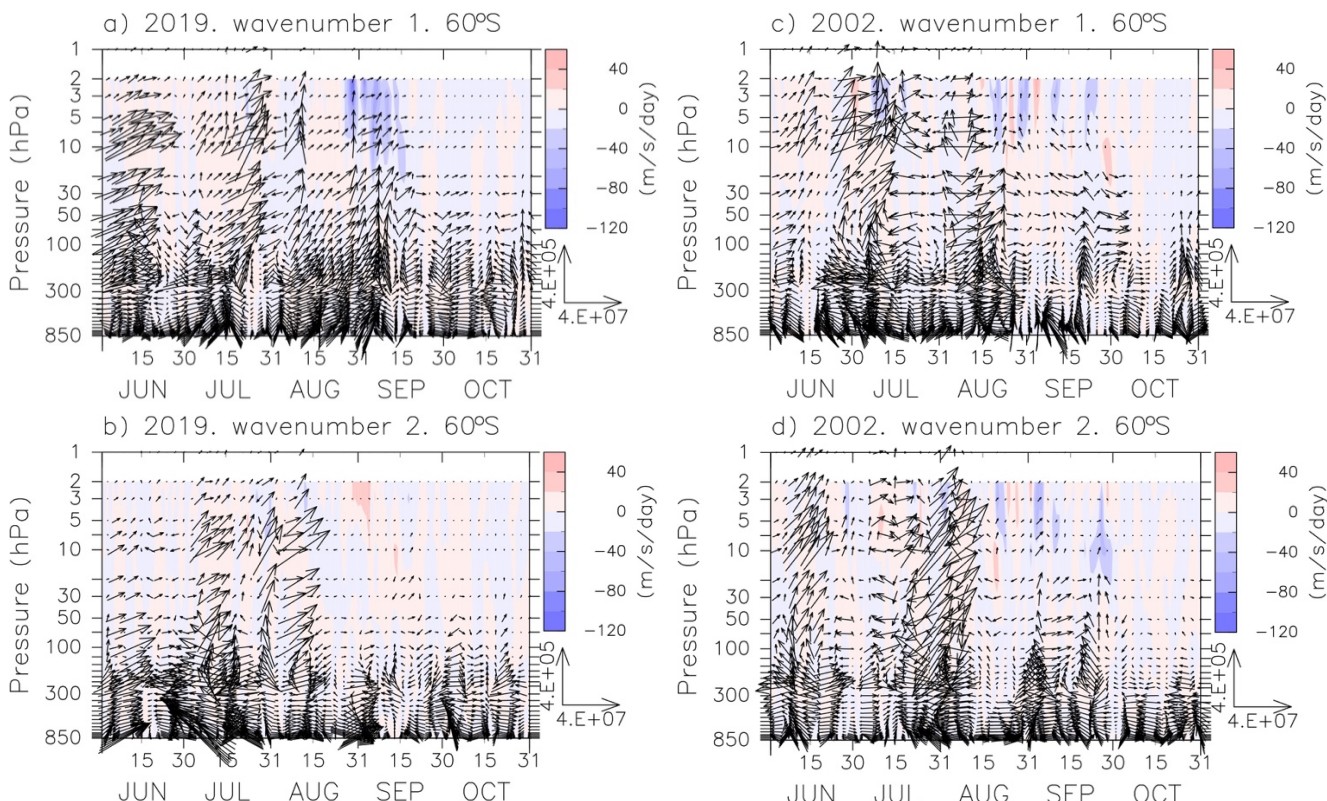

2  Figure 6. Time–height cross-sections of the E-P flux [kg s$^{-2}$] (vectors) at 60°S for zonal wavenumber 1 (a, c) and 2 (b, d) and

3  the wave driving due to its divergence [m s$^{-1}$ day$^{-1}$] (colour shading) from 1 June to 31 October in 2019 (left) and 2002 (right).

4  E-P flux vectors pointing to the right direction corresponds to the poleward. The blue (red) shading denotes the zonal wind

5  deceleration (acceleration).

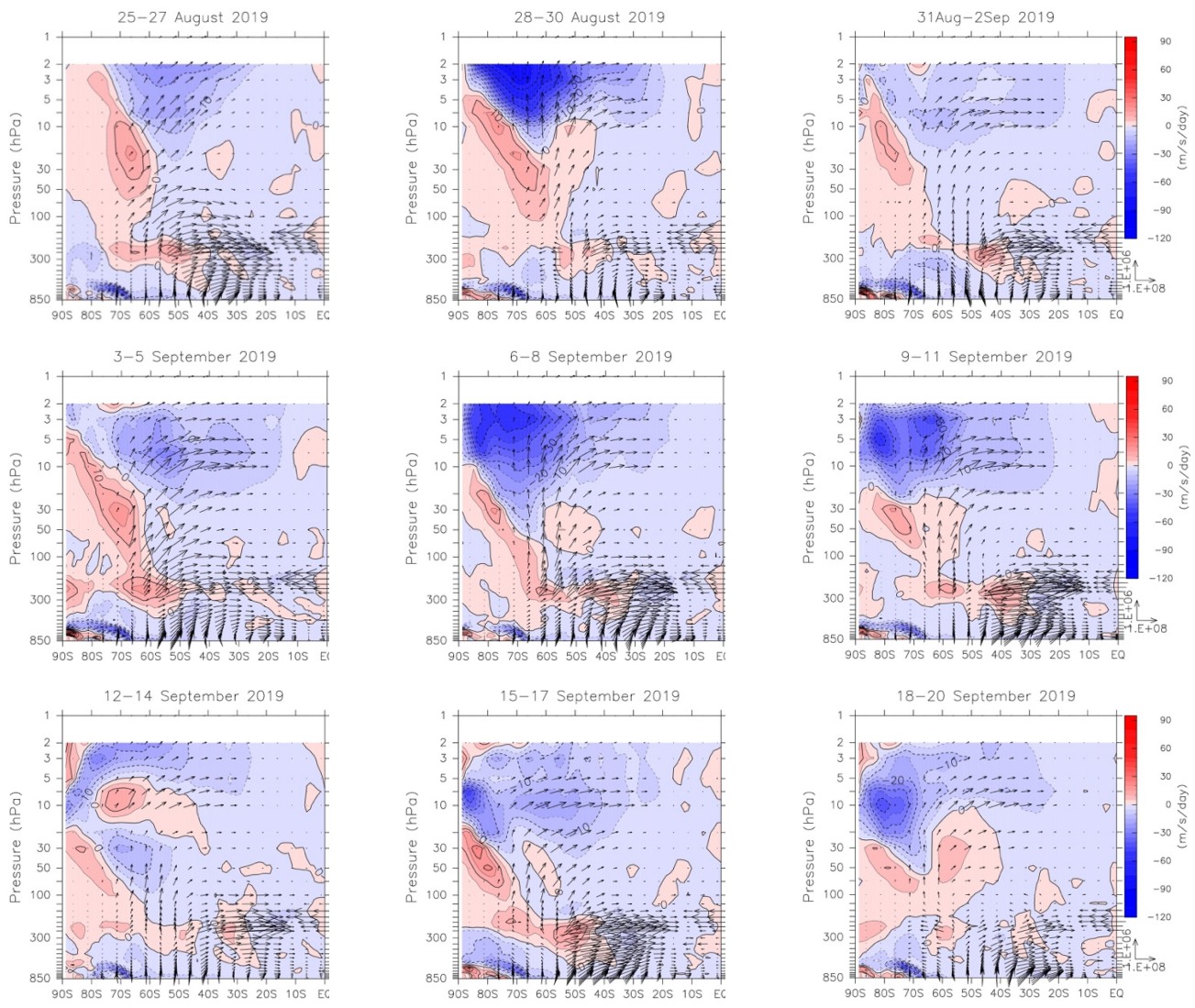

2    Figure 7. Same as Fig. 5 but for E-P flux [kg s$^{-2}$] (vectors) and the wave driving due to its divergence [m s$^{-1}$ day$^{-1}$] (colour

3    shading). Contour intervals are 5 m s$^{-1}$ day$^{-1}$.

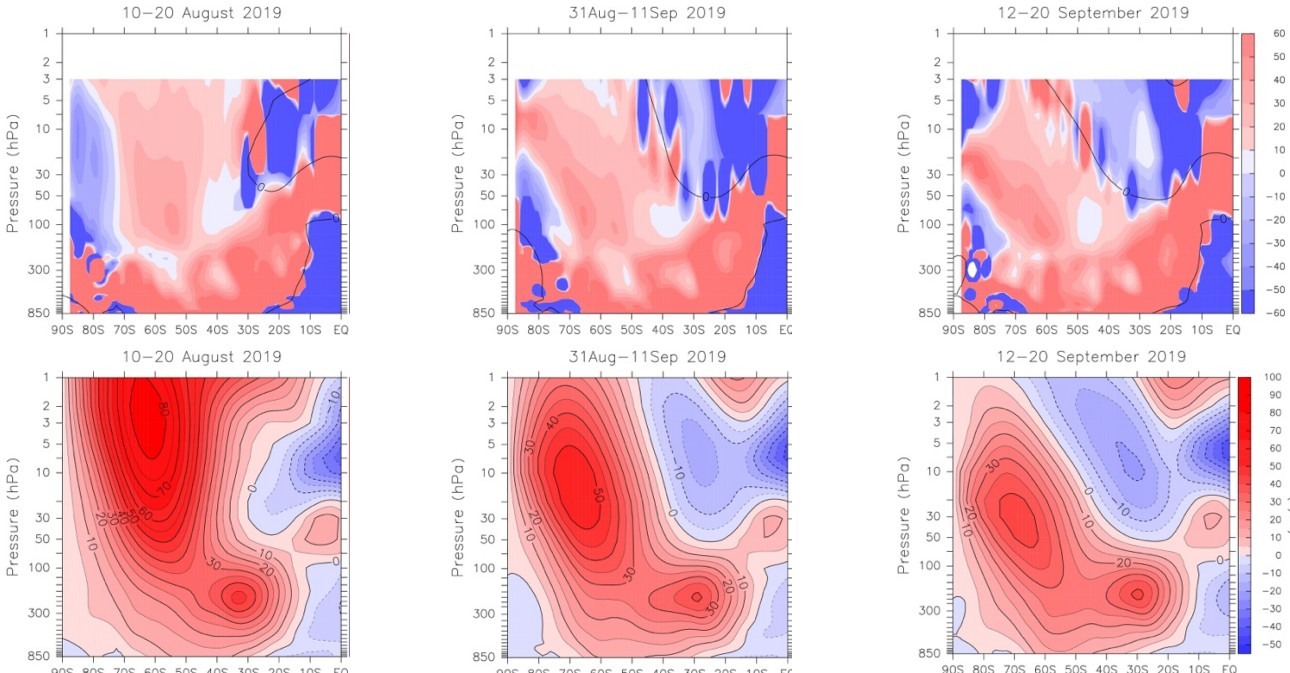

3  Figure 8. Same as Fig. 5 but for the quasi-geostrophic refractive index squared (dimensionless, colour shading, top) and zonal-

4  mean zonal wind ([m s$^{-1}$], bottom) averaged over 10–20 August (before the warming), 31 August–11 September (during the

5  warming), and 12–20 September 2019 (after the warming). Black lines in the top panels denote the zero-wind speed contour.

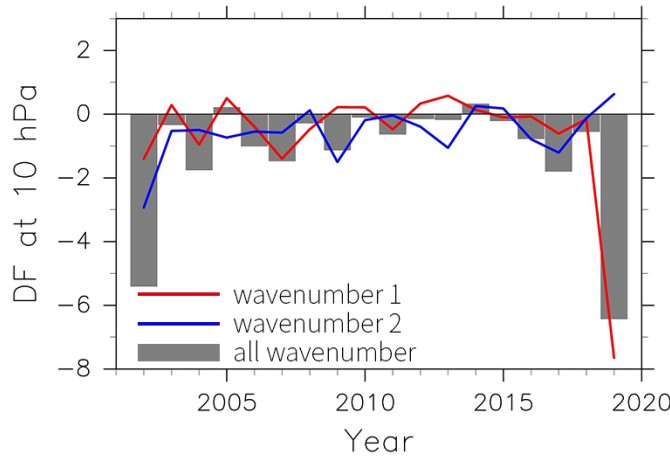

2 Figure 9. Interannual variations in wave driving due to E-P flux divergence [m s$^{-1}$ day$^{-1}$] averaged over 30°-90°S at 10 hPa in

3 September for zonal wavenumber 1 (red) and 2 (blue) from 2002 to 2019. Gray bars show the results for all wavenumbers.

