# Peer review of "Dynamical evolution of a minor sudden stratospheric warming in the Southern Hemisphere in 2019"

_Atmospheric Chemistry and Physics, 2021_

## Author Comment (AC1)

**Reply to the queries and comments of Referee #2**

We first thank the referee's efforts in considering our manuscript and making suggestions for improvement. In our detailed reply below, we reproduce the reviewer's comments in blue standard font, while our replies are in black standard font.

The paper provides what the title is implying (in the best sense of the words) – a description of the SSW in the SH during 2019. This might not sound much, but the authors try to provide a comprehensive comparison to the 2002 SSW (a more "severe" event) and they attempt to provide an explanation for how such events can develop in the SH – here, in particular a wave guide explanation is explored for the 2019 event and could presumably play a slightly more prominent role in the discussion to provide the paper with a more unique selling point. I do not have any mayor concern regarding the paper – however, it does need some revision (clarifications and twists) before final publication in ACP.

We appreciate the reviewer's positive overall response to our work and for the valuable suggestions provided. All referee's comments will be answered in detail when we submit a revised version of our paper.

Point-by-point – based on the abstract:

The paper uses JRA-55 from 2002 to 2019. Thus, the extremes "flank" the base-line analysis period. There should be an explanation why this is sensible – why not use earlier data as well?

Thank you for important comments about the abstract. We modified the starting year of analysis period from 2002 to 1979. Since there are limited observations in the upper stratosphere especially at high latitudes in the SH before 1979, we only use data from 1979-2019, at P3, L23. In addition, the satellite observations were included in the JRA-55 reanalysis data after 1979. The climatology mean is calculated over 41 years (1979-2019), at P3, L24-25.

We provide more material to explain the reason why we focus on the comparison between 2002 and 2019 in this study. Here we show Figure 1R below which is the same Figure as Figure 9 but with a long analysis period (1979-2019; please note we have changed the order of Figure 8 and 9 in the revision). In Figure R1 the values of planetary wave driving in 2002

and 2019 are much larger than other years in the analysis period. As this point, it is of high interest for comparing the SSWs in 2002 and 2019 in the SH.

[Figure]

Figure 1R: The same as Fig.9 but for from 1979 to 2019.

Phrases like "strong warming" should be avoided – "rapid temperature increase" (or similar) would be more appropriate.

We modified "strong warming" to "rapid temperature increasing" in the revision, at P1, L14. Thank you.

The way 2002 is compared with 2019 seems sometimes rather arbitrary. Maybe just state initially that 2002 has been explored extensively (in other papers) and summarise two key messages. Followed by a more complete/complex description of the 2019 event, in particular the unique role of PW1 and the waveguide point.

Accepted. We modified the order of the abstract following your advice, at P1, L10-16.

Point-by-point – based on the main text:

p1, l25: associated

Corrected, at P1, L28.

p1, l28: is this always causal? If the displacement starts first, the strengthening of the Z1/PW1 would happen without a source in the troposphere …

Thank you for the comment. We provided a precise description in the revision as follows. "The essential dynamical mechanism of the development of the SSWs is that enhanced quasi-stationary planetary waves propagate from the troposphere to the stratosphere and interact with the zonal mean flow (Matsuno, 1971).", at P1, L31-32.

p1, l31: weak wave activity is ambiguous – the definition should be clearer …

We modified the definition to clarify this as follows. "One of the reasons that SSWs rarely occur in the SH is the distribution of ocean–land and orography, which leads to smaller planetary wave amplitude in the SH (Andrews et al., 1987; Newman and Nash, 2005).", at P2, L2-4.

p2, l8: preconditioning should be briefly explained (beyond the citation) …

We provided explanations about preconditioning in the revision as follows. "It has been reported that minor SSWs characteristically precede major SSW as "preconditioning". The preceding minor SSWs are associated with planetary waves amplification of zonal wavenumber 1 concurrently with a minimum of the zonal wavenumber 2 (Labitzke1977; Labitzke1981; Bancalá et al., 2012). The "preconditioning" also changes in the zonal flow that weakens the polar night jet and thus favors the upward and poleward propagation of planetary waves (Andrews et al., 1987; Labitzke, 1981; Manney et al., 2009). Following the poleward propagating planetary waves, the polar vortices become vulnerable that lead to the causing major warmings. The presence of precondition is a necessary condition for a major SSW to occur but not a sufficient condition (Limpasuvan et al., 2004).", at P2, L13-19.

p3, l1: plausible factors – what does this mean? A proposed mechanism for the initiation of the SSW in 2019? Please explain!

We intend to give possible factors that may explain the cause of SSW2019. We have rewritten the last paragraph in the introduction section in the revision, at P3, L14-17.

p3, l6: uses

Corrected, at P3, L21.

p3, l7: Please provide a rational for the time period chosen (see above)!

We have used the period from 1979 to 2019 for climatological analysis, at P3, L23. In addition, the following sentence was added to the current manuscript, "The climatological mean is calculated over 41 years (1979-2019).", at P3, L24-25.

As we explained above in Figure R1. As the extraordinary features of planetary wave driving in 2002 and 2019 during the analysis period (1979-2019), we believe the study of the two years can help better to understanding the SSW in the SH.

p3, l14: employ; What kind of wave forcing? Presumably some kind of "effective wave forcing" as the quantities are calculated from the gridded data and not the tendencies in the model - already daily averages are used?

Thank you for the comments. Corrected (P3, L31); The wave forcing here is referred as the forcing by planetary waves with zonal wavenumbers from 1 to 3 (the planetary scale) that could affect the zonal-mean circulation in the stratosphere (P3, L30-31); We used daily averaged gridded data and not the tendencies in the model.

p3, l21: steady zonal flow? Is this the zonal mean of quantity X (zonal wind, geopotential, etc.) based on the daily averaged data?

Corrected, at P4, L5. Yes, we refer to zonal-mean zonal wind, geopotential, etc based on the daily averaged data.

p3, l26: presumably "u overbar" is the zonal mean zonal wind (and not the horizontal basic flow)?

"u over bar" represents the zonal-mean zonal wind. Corrected in the revision, at P4, 11-12. Thank you.

p3, l29: explain briefly the Fourier transform and how you will derive the fluxes associated with the waves (basically slightly more detail would be nice) …

Thank you for the comment. We provide more details about the Fourier transformation used in our study in the revision, at P4, L5-7.

p4, l1: Overview of SSW 2019 – I would suggest an overview of the 20002 event first (as the baseline case), followed by a description of the 2019 event – also it would be nice to provide a direct comparison of the temporal evolution (timing) of the warmings – the figures exist, but the logic is not quite obvious to me … (thus the suggestion to start with 2002, the literature and your descriptive plots of 2002 and to contrast those facts with 2019 and afterwards detailing 2019 …)

Thank you for the comment. We modified the paragraph following your suggestion in the revision, at P4, L17- P5, L6.

p4, l8: what is "pronounced high temperature"?

Here we refer to high temperature at the South Pole than that in 60S (positive $\Delta T$) in from late August to early September. We revised the description in the revision, at P4, L30.

p4, l26: warm air becomes warmer sounds too colloquial – presumably this could be phrased differently.

We modified the phrase following your suggestion, at P5, L12.

p4, l28: presumably you could explain why this is obvious – in the current structure wave amplitudes are introduced in figure 4 (much later) …

We have deleted the "Corresponding to the amplified PW1" in the revision to avoid misunderstanding, at P5, L12. Thank you.

p5, l2: "daily changes" sound like a derivative … the figure shows a time series of daily data

Thank you for the comment. The word "daily" has been deleted. We revised the sentence as follows in the revision, "Figure 3a shows zonal-mean temperatures at 90°S and 10 hPa from 1 June to 31 October.", at P5, L20.

p5, l14: I don't understand the second part of the sentence – the chosen location is representative of the PNJ (close to the climatological maximum of the PNJ)?

In order to avoid misunderstanding, we have deleted the second part of the sentence in the revision, at P5, L30. As mentioned in the introduction (P2, L6-10), we followed the definition of a major or minor SSW that whether a reversal of zonal-mean zonal winds occurs at 60S and 10 hPa.

p5, l28: "daily changes" sound like a derivative … the figure shows a time series of daily data

The words "daily changes" has been deleted in the revision. The revised sentence is as follows. "Figure 4 shows planetary wave amplitudes of PW1 and PW2 at 60°S and 10 hPa (top) and upward E–P fluxes for PW1–3 at 60°S and 50 hPa (bottom) for 2019 (left) and 2002 (right).", at P6, L10-11. Thank you.

p5, l33: Clear phrasing: PW1 and 2 refer to the actual wave and Z1 and Z2 are the corresponding amplitudes (the wave phase is not discussed – only indirectly with the fluxes) …

We modified the phrase following your comments, at P6, L13-14. Thank you.

p6, l20: "high latitudes of the stratosphere" is not very precise, please provide a latitude and altitude range …

We modified the phrase "high latitudes of the stratosphere" to "about 60°S and in the range from 5 to 1 hPa" in the revision, at P6, L34. Thank you.

p6, l32: what is actually propagating? Presumably one would distinguish features that apparently "move" and waves that propagate – just a note of caution …

Thank you for the comment. As previous studies show that the E-P flux is a useful diagnostic for evaluating the propagation of waves in the stratosphere. The E-P flux vectors represent the direction of planetary wave group velocity, and the divergence (convergence) relates to the acceleration (deceleration) of the westerly zonal-mean zonal wind. Here we use the E-P flux to diagnose planetary wave propagation in the stratosphere. We have restructured the paragraph in the revision, at P7, L13-16.

p7, l1: be more precise – what do you call wave activity in this context? (This is a request to be more precise when referring to the quantities in the figures …)

We modified the sentence as follows "In contrast to the PW1, the PW2 is relatively weak during the warming period (Fig. 6b)." in the revision, at P7, L17-18. The word "activity" has been deleted to avoid misunderstanding.

p7, l4: as the point before – and: is there a corresponding plot in a "proper" publication?

We thank you for your comments. Baldwin et al. (2003) present the evolution of planetary wave activity in SSW2002. We cited the reference in the revision, at P7, L21. The new reference has been added in the reference list, at P13, L13-14.

p7, l25: More framing would be nice – this is actually the best part of the paper. Do provide the reader with a better feeling for your hypothesis!

We appreciate your helpful comment. We added the following sentences at P9, L2-7.
"Even though SSW2019 did not fulfil the criterion of a major SSW, the large increasing temperature in high latitudes still has a significant impact on the stratosphere. Due to the remarkable increased polar stratospheric temperature, slowing down catalytic chemical reaction on polar stratospheric clouds that suppress the formation of the Antarctic ozone hole in austral spring. Indeed, a diminished Antarctic ozone hole area is observed in 2019 (Wargan et al., 2020; Safieddine et al. (2020). As described in previous section, SSW2019 resulted from the pronounced planetary wave forcing especially the contribution from zonal wavenumber 1. In this section, we consider such unusual features in SSW2019 and compare it with SSW2002."

 Figure 8 actually shows a time series – from this time series the reader can take (with your help) information regarding the interannual variability and the unique characteristics of 2002 and 2019 …

Thank you for the comment. We modified the description of Figure 9, at P8, L27-28. Please note that we have changed the order of Figure 8 and Figure 9 in the revision.

p8, l2: If you would provide more information about the concept of preconditioning earlier (as suggested above), you could come back to it here and provide the case specific detail in more depth.

Thank you for the comment. We added one paragraph to provide description on preconditioning in the introduction section (P2, L13-19).

p8, l7: I was wondering if the eastward travelling wave point would merit some more detail … (depends a bit on how the preconditioning will be explained in more detail)

Thank you for the comment. We cited the studies of Krüger et al. (2005), which discussed the importance of eastward-traveling PW2 in 2002. We provide results of the eastward-traveling PW2 in 2019, at P6, L17-19. A discussion of eastward-travelling PW2 in 2019 are made, at P9, L14-17.

p8, l15: what is meant with "abrupt occurrence"? An increase in amplitude (in conjunction with the preconditioning)?

Because of the comment on the structure of discussion section, we have totally restructured the discussion section. Please note that the paragraph involved the phrase "abrupt occurrence" has been deleted in the revision in order to avoid misunderstanding. In the previous version, we intended to express the occurrence of the increased amplitude of planetary wave which begins in late August in 2019.

p8, l17: I do not disagree, but the mechanism as such should be explained in more detail …

Thank you. We have added one paragraph to carefully discuss the refractive index squared in SSW2019, at P10, L4-13. Please note that we have restructured the discussion section in the revision, the results on the refractive index squared are presented in P8, L8-20.

p8, l28: This seems a central point. Alas, the current phrasing is not very clear and some more explanation and a better description would be nice. This seems to be the "nugget" of the paper and it would be nice to give it the prominence it deserves. Also, how does this compare to 2002? Or is a direct comparison (due to the different timing of the events) not possible? A comment along those lines would be useful.

Thank you very much. As the mentioned above, in order to provide clear explanation of the "nugget" of our paper, we have separated the results (P8, L8-20) and discussion (P10, L4-13) of the refractive index squared for SSW2019.

We compared the refractive index squared for 2002 here (Figure 2R, 3R) as supply material, but not in the manuscript. As you said, due to the different timing of the events, we provide monthly mean latitude-height cross section in July, August, and September. Firstly, the positive values of refractive index squared in August and September 2019 around 60S from lower to higher stratosphere are large. This provides a better environment for propagation of planetary waves into the stratosphere in 2019. Furthermore, an apparent poleward shift of the positive refractive index squared is found in September 2019. Large positive values of the refractive index squared are found in the stratosphere, but similar shifts towards the pole is not clear in 2002.

p8, l32: The "Summary and Conclusion" needs adjusting to the changes triggered by the above …

We have restructured the summary and conclusion section according to the changes in the previous sections, at P10, L30-P11, L30. Thank you.

Figure 1: Do provide coherent labelling (delta T <-> T gradient) – the gradient is actually not a gradient – it is a temperature difference between two latitudes – please remind the reader how JRA provides the zonal mean at the pole … (if possible, please add units to the colour bars)

Corrected. The grid values at the Pole in JRA-55 contains the same value with the same format as in other latitudes with the same grid number in East-west (zonal) direction. Thank you.

Figure 2: You average three days (averaged from 6 hourly data), and you do this subsequently for all three-day periods within your analysis period – "every three days" is also "over three days" – just make sure the description is as precise as possible … (if possible, please add unit to the colour bar)

"Every three days" is modified to "for successive 3-day mean" in the revision, at P18, L3. Unit are added in the color bar. Thank you for your comment.

Figure 3: Please include a colour key in the figure (just describing it in the text is prone to error). Aside: This figure is obviously useful to mention the timing of the warmings, when discussing it.

Corrected.

Figure 4: y-labels for the bottom plots are missing, colour keys should be provided in the plot, presumably clarity of the description could be improved by avoiding the "verbal colour key".

Corrected.

Figure 5: Please see comment for figure 3; labels at y-axis, if possible …

Corrected.

Figure 6: Labels at y-axis, if possible; unit at colour bar, if possible … (Why WN here? Everywhere else PW …)

Corrected.

Figure 7: Labels at y-axis, if possible

Corrected.

Figure 8: Time series of … (see comment above); in September – exact definition, please; colour key in the plot would be nice …

Thank you for the comments. Color key is added.

The reason that we use the wave driving due to E-P flux divergence in September is that both in 2002 and 2019 the planetary wave become active in September which could be seen in Figure 4. Furthermore, as we also added the following sentence in the revision to explain the use of September. "Because upward propagation of planetary waves from the troposphere to the stratosphere develop from July and peaks in September (Lim et al., 2021).", at P8, L22-24.

As we have explained in Figure R1, compared with other years the extraordinary features of planetary wave driving can be observed in 2002 and 2019. In order to display clearly, we chose the short time period (2002-2019) figure to present.

Figure 9: Labels at y-axis, if possible

Corrected. Thank you.

[Figure]

[Figure]

Figure 2R: The same as Fig.8 but for July, August, and September 2019.

[Figure]

Figure 3R: The same as Fig.2R but for July, August, and September 2002.

[revised manuscript text omitted]

---

## Author Comment (AC2)

**Reply to the queries and comments of Referee #3**

We first thank the referee's efforts in considering our manuscript and making suggestions for improvement. In our detailed reply below, we reproduce the reviewer's comments in blue standard font, while our replies are in black standard font.

In their paper, Liu et al. present an analysis of the 2019 minor SSW in comparison with the 2002 major SSW. The paper is well written and nicely describes the dynamical situation of the 2019 SSW. Unfourtunately, the study is not very thorough and does not provide many new insights. The main result that the wave guide may be the decisive feature for the generation of the SSW is interesting, but neither pointed out well enough in the paper, nor it is analysed in any detail (see below in the major points). Furthermore, I think some restructuring of the paper is required and possibly also some compressing, as a few points are repetitive. In general, I agree with reviewer #1 who writes that the paper can be published, but extensive revision should be made before that. I want to add a few points to those of reviewer #1, though, please see below.

We appreciate the reviewer's valuable suggestions provided. All the comments will be considered in detail when we submit a revised version of our paper.

On a different page I want to make the authors, but also the ACP editorial aware of the fact that for these type of purely dynamical studies there is since a good while the new copernicus journal Weather and Climate Dynamics (WCD). I don't think that journal stands in competition to ACP, rather it is complementary and in my opinion better suited for papers like this one, which actually does not include any chemical analysis and the physics is largely limited to dynamics. My point is, if my thinking on this is correct, the editors should probably deflect these type of papers from ACP to WCD.

We would like to leave it to the editor's judgment, but the authors think that ACP has more readers and dynamic discussion is also possible. As this topic is of high interest for the ozone hole evolution in the SH, we believe it fits better to ACP.

Major issues:

All in all, I think not all figures add significant value to the story. E.g. Fig. 5 does not really provide more information than what had already been shown in Fig. 4 and moreover, some

of the panels are more or less repeated in Fig. 9. So I'd say that one is obsolete. I have a similar feeling about Fig. 7, which does not really add anything to what had already been told in Fig. 4 and Fig. 6.

Thank you for instructive comments on the paper and figures. We think that detailed information on latitude-height sections of zonal-mean zonal winds and wave driving as presented are necessary for understanding the characteristics of the SSW2019.

I think the authors should go through the paper again and reconsider what is really necessary to tell the story and what is maybe only a side note. Section 4 "Discussion" needs to be seriously restructured. Only the middle part of it actually is a discussion (P8L1-13). The beginning and the end are further results parts and should treated as such. Instead, some of the real discussion seems to be in Sect. 5, e.g. P9L26-31.

Thank you for instructive comments. We reconsidered and restructured the discussion section, at P9, L2-P10, L27. All the analysis results are provided in the results section, at P4, L17-P8, L33.

The paper "only" compares the two SSW events. In the SH, several other minor SSWs have taken place, mainly in the early 2000s and in the 80s. Why do you not compare the 2019 event to those as well?

This is a very good comment. The main reason that we only compare the SSWs in 2019 and 2002 in this study is that SSW2002 is the only observed major SSW ever in the SH, even though the SSW2019 could be regarded as a minor SSW, but with a large impact. It led to an anomalous small ozone hole and significant reduction of the ozone total column in 2019 (Safieddine et al., 2020; Wargan et al., 2020). The classification of the SSW2019 is important, but also the impacts on the stratosphere. However, the other minor warmings in the SH did not bring such impact on the stratosphere as SSW2019 did. Here we provide more material to explain, Figure R1 shows the same figure as Figure 9 including the analysis period (1979-2019) below, showing also other events for the analysis period. We use the data after 1979 because of there are limited observations at high latitudes in the SH before that year. In addition, the satellite observations were included in the JRA-55 reanalysis data after 1979. The big difference in the values of planetary wave driving between 2002/2019 and other 39 years can easily observe. As this point, it is of high interest for comparing the SSWs in 2002 and 2019 in the SH.

[Figure]

Figure 1R: The same as Fig.9 but for from 1979 to 2019.

Moreover, you give a quick reference to the S-RIP data. Can you elaborate on how clear and robust your findings are with regard to your JRA-55 data set. Are the results similar in other reanalysis data sets?

Kawatani et al. (2016, ACP, https://doi.org/10.5194/acp-16-6681-2016) and Kawatani et al. (2020, ACP, https://doi.org/10.5194/acp-20-9115-2020) compared various reanalysis data for the presentation of the QBO and SAO in the connection with the S-RIP and showed that the standard deviation of zonal wind and temperature fields among reanalysis data was overall small throughout the stratosphere except near the Equator. Thus, the other reanalysis data are expected to give similar results to the current analysis.

In particular, as the wave guide is your main result, that should probably be consolidated with at least one other reanalysis data set or a model that reproduces the minor 2019 SSW. The study ends just where things start to become intersting, this is very disappointing. It is interesting how the different waves act together in 2002 and they don't in 2019. It is obvious that wave generation and wave propagation must be analysed in the next step.

Thank you for your comment on the waveguide that we discussed in the paper. Further analyses are needed, for example the origin of the planetary wave as you have mentioned. In Yamazaki et al. (2020), they suggest the possible origins of the planetary waves, we also

provide our considered future analyses for SSW2019 in discussion section (P10, L25-27). We agree with your comment on using a different dataset to make a comparison in the future study.

Special situations for wave generation have not been mentioned at all. The wave guide change is a nice result, but now it would be interesting how and why the refractive index (RI) forms this way. To my knowledge, the RI mainly depends on winds and temperatures. So can the preconditioning with the strong winds be responsible for the way the wave guide forms? Is wave forcing prior to the SSW event responsible for the formation of that? If the authors refuse to make more analyses on this, I would at least expect some discussion and speculation in this direction, such that this study can be taken up as a starting point for a deeper analysis.

Thank you for positive comment on the result of refractive index square. Because of the insufficient explain in the paper, the special structure of refractive index square in 2019 results from the zonal-mean zonal winds structure as what you mentioned. That is the exact reason why we provide the latitude-height section of zonal-mean zonal winds during the same time periods in the bottom in Figure 8. We followed your comment on restructuring the discussion and summary/conclusion sections. In the revision, we present all the results of the refractive index square in P8, L8-20. We also provide a new paragraph for discussion of refractive index square in SSW2019, at P10, L4-13.

And that brings me to my last major point:The paper does not provide any implications or outlook, or ideas for further studies how to get deeper into understanding this and especially, how to use your results to improve predictability on S2S time scales. Can for example the wave guide be predicted, are there any implications? A discussion on that should round up the paper in my eyes

Predictability about the SSW2019 on the S2S models is a nice comment. We mentioned the related study by Rao et al. (2020) in the introduction section (at P3, L2-5). In the current study, we focus on analyzing the SSW2019 in the reanalysis data set and compare it with the only major SSW observed in the SH. Implications like predictability of SSW2019 on S2S is beyond the scope of the current paper, that will be the near future study.

Minor and technical issues:

• P1L16: With "the values are larger", do you mean the wave driving was stronger? If so, write it.

Thank you for the comment. Here "the values are larger" refer to the planetary wave driving was stronger. We rephrased it in the revision, at P1, L19.

P1L24 remove "hereafter referred to as"

Removed. Thank you.

P1L25 associated

Corrected, at P1, L28. Thank you.

P1L26 during an SSW

Corrected, at P1, L29. Thank you.

P1L17-19: This sentence is not really comprehensible this way when you are not already clear about the situation. Hence, please rephrase it.

Thank you for the comment. We rephrased it in the following way. "Major SSWs tend to accompany preceding minor warmings, preconditioning, which changes the zonal flow that weaken the polar night jet as seen in SSW2002. A similar preconditioning was hardly observed in SSW2019.", at P1, L20-21.

"the ocean-land and orography distribution", plus, remove "and small wave perturbations"

Corrected, at P2, L2-4. Thank you.

It looks to me like the recent literature on the topic has not fully been addressed. I think for example the studies by Lee et al. (2020) and by Shen et al. (2020) have dealt with the topic too and their results could add value to this one (10.1029/2020JA029094 and 10.1029/2020GL089343)

Thank you for the comments. We added relevant literatures including Shen et al. (2020) and Lee et al. (2020) in the introduction section in the revision, at P3, L8-12. These references have been added in the reference list, at P14, L28-31; P15, L22-23.

P5L3: change "normal" to "average"

Corrected, at P5, L19. Thank you.
In P8L17-19 you state that you provide a plausible mechanism (the wave guide), but you provide the mechanism only afterwards. Turn that around!

Thank you for comment. We restructured the whole discussion section in the revision (P9, L2- P10, L27). The results of refractive index square (the waveguide) are presented in the results section (P8, L8-20). We added one paragraph for the discussion of the refractive index square in the discussion section (P10, L4-13).

On P8L17-18 the sentence "wave guide propagation from the troposphere to the strato- sphere is controlled by the index of refraction" is written as if this would be a new result.

But to that end this is just common theory. Please rephrase this.

Corrected. We rephrased the discussion section in the revision (P10, L4-13). We deleted "wave guide propagation from the troposphere to the stratosphere is controlled by the index of refraction" as this is a common theory in the revision. Thank you.

P5L28 Figure 4 shows time series of daily data of the geopotential....

The fact that you see daily changes of PW1 and PW2 amplitudes can be put below in L31 or so.

Corrected, at P6, L10-11. Thank you.

Add labels to the colour bars at figures 1, 2, 5, 6, 7, 9

Corrected. Please refer to the new figures in the revision. Please note that we changed the order of Figure 8 and 9 in the revision. Thank you.

Add legends to fig. 3, 4, 8

Corrected.

Corrected, at P3, L21.

Fig.1: add "â ¦" to the 60S in the titles

We cannot understand your comment because of the unclear display. Presumably your comment is that change 60S into "60°S". We changed 60S to "60°S" in the revision.

P18L3: What amplitude?

We modified "amplitude" to "planetary waves amplitudes", at P20, L4.

Fig.6: Bad choice of colour bar values. Rather choose smaller max and min values, such that something can actually be seen here.

We thank you for the comment. We modified the color bar and its values in the revision, at P22.

[revised manuscript text omitted]